# How does the terrestrial carbon exchange respond to interannual climatic variations? A quantification based on atmospheric $CO_2$ data

**Christian Rödenbeck[1], Sönke Zaehle[1], Ralph Keeling[2], and Martin Heimann[1,3]**

[1]Max Planck Institute for Biogeochemistry, Jena, Germany
[2]Scripps Institution of Oceanography, University of California, San Diego, USA
[3]Institute for Atmospheric and Earth System Research (INAR), Faculty of Science, University of Helsinki, Finland

*Correspondence to:* C. Rödenbeck (christian.roedenbeck@bgc-jena.mpg.de)

**Abstract.** The response of the terrestrial Net Ecosystem Exchange (NEE) of $CO_2$ to climate variations and trends may crucially determine the future climate trajectory. Here we directly quantify this response on interannual time scales, by building a linear regression of interannual NEE anomalies against observed air temperature anomalies into an atmospheric inverse calculation based on long-term atmospheric $CO_2$ observations. This allows us to estimate the sensitivity of NEE to interannual variations in temperature (seen as climate proxy) resolved in space and with season. As this sensitivity comprises both direct temperature effects and effects of other climate variables co-varying with temperature, we interpret it as "interannual climate sensitivity". We find distinct seasonal patterns of this sensitivity in the northern extratropics, that are consistent with the expected seasonal responses of photosynthesis, respiration, and fire. Within uncertainties, these sensitivity patterns are consistent with independent inferrences from eddy covariance data. On large spatial scales, northern extratropical as well as tropical interannual NEE variations inferred from the NEE-T regression are very similar to the estimates of an atmospheric inversion with explicit interannual degrees of freedom. The results of this study offer a way to benchmark ecosystem process models in more detail than existing effective global climate sensitivities. The results can also be used to gap-fill or extrapolate observational records, or to separate interannual variations from longer-term trends.

## 1 Introduction

About a quarter of the carbon dioxide ($CO_2$) emitted to the atmosphere by human fossil fuel burning and cement manufacturing is currently taken up by the terrestrial biosphere (Le Quéré et al., 2016), thereby slowing down the rise of atmospheric $CO_2$ levels and thus mitigating climate change. The magnitude of this terrestrial Net Ecosystem Exchange (NEE) of $CO_2$, however, is subject to substantial variability and trends, to a large part as a response to variations and trends in climate. Due to this feedback loop, the response of NEE to climate may crucially determine the future climate trajectory (Friedlingstein et al., 2001), yet present-day coupled climate–carbon cycle models strongly disagree on its strength (Friedlingstein et al., 2014).

To reduce these uncertainties, observations of present-day year-to-year variations have been used as a constraint on the unobservable longer-term changes (Cox et al., 2013; Mystakidis et al., 2017), using the finding that these models show a close link between the climate–carbon cycle responses at year-to-year and centennial time scales. It cannot be known, however, to which extent this link indeed holds in reality (Mystakidis et al., 2017). While carbon cycle anomalies on the year-to-year time scale are clearly attributable to climate anomalies (through the variable occurrance of sunny/cloudy, warm/cold, wet/dry days or periods), additional longer-term trends may arise as a response to growing nitrogen and $CO_2$ fertilization, to slow warming, to expanding or shrinking vegetation, to adaptation of ecosystems, to shifts in species composition, or to changing human agricultural practices and fire suppression. Some of these processes may also slowly change the strength of the

short-term climate–carbon cycle responses over time. More-over, both year-to-year and decadal/centennial carbon cycle changes are overlaid by the much larger periodic variability (day/night cycle, seasonal cycle). When using observations to constrain the climate–carbon cycle responses, therefore, it is essential to employ observational records spanning time periods as long as possible to get statistically significant results, and to separate the signals on seasonal, interannual, and decadal time scales (compare Rafelski et al., 2009).

Variability and trends of terrestrial carbon exchange have been observed through a variety of sustained measurements, including local measurements by eddy covariance towers measuring ecosystem fluxes (e.g., Baldocchi et al., 2001; Baldocchi, 2003) and indirect measurements by satellites recording changes in vegetation properties (e.g., Myeni et al., 1997). The longest observational records are the atmospheric $CO_2$ measurements, started in the late 1950s at Mauna Loa (Hawaii) and South Pole by Keeling et al. (2005) and since then extended into a network of more than 100 $CO_2$ sampling locations worldwide. Based on the Mauna Loa long-term record considered to reflect global $CO_2$ fluxes, a close link between atmospheric $CO_2$ growth rate and tropical temperature variations has been established (e.g., Wang et al., 2013). Using measurements from Barrow (Alaska) conceivably reflecting variations in boreal $CO_2$ fluxes, similar relationships have been suggested for high-latitude ecosystems (e.g., Piao et al., 2017).

Extending these analyses, the aim of this study is to directly quantify the contributions of the different seasons and different climatic zones to the response of NEE to interannual climatic variations, in order to obtain more process-relevant information. To this end, we combine a linear regression between NEE and climate anomalies with an "atmospheric inversion" (e.g., Newsam and Enting, 1988; Rayner et al., 1999; Rödenbeck et al., 2003; Baker et al., 2006; Peylin et al., 2013) which quantitatively disentangles the atmospheric $CO_2$ signal into its contributions from the various regions and times of origin, and allows us to make use of multiple long-term atmospheric $CO_2$ records. In addition to the atmospheric data, eddy covariance data are used for independent verification.

## 2  Method

### 2.1  The standard inversion

As a starting point, we use the existing Bayesian atmospheric $CO_2$ inversion implemented in the Jena CarboScope, run s85oc_v4.1s (update of Rödenbeck et al., 2003; Rödenbeck, 2005, see http://www.BGC-Jena.mpg.de/CarboScope/). It estimates spatially and temporally explicit $CO_2$ fluxes between the Earth surface and the atmosphere, based on atmospheric $CO_2$ measurements from 23 stations (marked with * in Table 1) each of which spans the entire analysis period

(chosen here to be 1985-2016 when more data are available, see Rödenbeck et al. (2018) for runs over 1957-2016). Using an atmospheric tracer transport model to simulate the atmospheric $CO_2$ field that would arise from a given flux field, the inversion algorithm finds the flux field that leads to the closest match between observed and simulated $CO_2$ mole fractions. In addition, the estimation is regularized by a-priori constraints meant to suppress excessive spatial and high-frequency variability in the flux field. The a-priori settings do not involve any information from biosphere process models. Fossil fuel fluxes are fixed to accounting-based values. In the particular run s85oc_v4.1s used here, ocean fluxes are fixed to estimates based on an interpolation of surface-ocean $pCO_2$ data (Jena CarboScope run oc_v1.5). A more detailed technical specification, including references and highlighting changes with respect to earlier Jena CarboScope versions, is given in Appendix A.

For reference in Sect. 2.2 below, we mention here that this standard inversion calculation represents the total surface-to-atmosphere $CO_2$ flux $\mathbf{f}$ as a decomposition

$$\mathbf{f} = \mathbf{f}_{\text{NEE,LT}}^{\text{adj}} + \mathbf{f}_{\text{NEE,Seas}}^{\text{adj}} + \mathbf{f}_{\text{NEE,IAV}}^{\text{adj}} + \mathbf{f}_{\text{Ocean}}^{\text{fix}} + \mathbf{f}_{\text{Foss}}^{\text{fix}} \quad (1)$$

into adjustable long-term mean terrestrial NEE ($\mathbf{f}_{\text{NEE,LT}}^{\text{adj}}$), adjustable large-scale seasonal NEE anomalies ($\mathbf{f}_{\text{NEE,Seas}}^{\text{adj}}$), adjustable interannual and shorter-term NEE anomalies ($\mathbf{f}_{\text{NEE,IAV}}^{\text{adj}}$), the prescribed ocean fluxes ($\mathbf{f}_{\text{Ocean}}^{\text{fix}}$), and the prescribed fossil fuel emissions ($\mathbf{f}_{\text{Foss}}^{\text{fix}}$). All these terms represent spatio-temporal fields.

This standard inversion will be used as a reference to compare the results of the NEE-T inversion introduced below (Sect. 2.2) at large spatial scales. Further, we used its estimated NEE variations in preparatory tests to confirm that NEE-T correlations actually exist, and to determine the degrees of freedom needed to accomodate their spatio-temporal heterogeneity.

### 2.2  The NEE-T inversion

Compared to the standard inversion (run s85oc_v4.1s), the NEE-T inversion (base run s04XocNEET_v4.1s) uses the same transport model and the same prescribed data-based $CO_2$ fluxes of the ocean ($\mathbf{f}_{\text{Ocean}}^{\text{fix}}$) and fossil fuel emissions ($\mathbf{f}_{\text{Foss}}^{\text{fix}}$). It also possesses the same adjustable degrees of freedom representing the long-term mean $CO_2$ fluxes (term $\mathbf{f}_{\text{NEE,LT}}^{\text{adj}}$) and its large-scale seasonality ($\mathbf{f}_{\text{NEE,Seas}}^{\text{adj}}$).

The NEE-T inversion differs only by replacing the explicitly time-dependent interannual NEE variations ($\mathbf{f}_{\text{NEE,IAV}}^{\text{adj}}$) with a linear NEE-T regression term plus residual terms,

$$\mathbf{f}_{\text{NEE,IAV}}^{\text{adj}} \quad \rightarrow \quad \gamma_{\text{NEE-T}}^{\text{adj}} w(\mathbf{T} - \mathbf{T}_{\text{LT+Seas+Deca+Trend}}) \quad (2)$$
$$+ (1-w)\mathbf{f}_{\text{NEE,IAV}}^{\text{adj}} + \mathbf{f}_{\text{NEE,Trend}}^{\text{adj}} + \mathbf{f}_{\text{NEE,SCTrend}}^{\text{adj}}$$

$\mathbf{T}$ represents the monthly spatio-temporal field of air temperature, taken from GISS (Hansen et al., 2010;

GISTEMP Team, 2017), interpolated to the spatial grid and daily time steps of the inversion (Appendix A). Its long-term mean, mean seasonal cycle, and decadal variations including linear trend ($\mathbf{T}_{\mathrm{LT+Seas+Deca+Trend}}$) have been subtracted to only retain interannual (including non-seasonal month-to-month) anomalies. The scalar $w$ is a temporal weighting being 1 within the analysis period 1985-2016 and zero outside; this ensures that the regression is specifically referring to this period. The interannual temperature anomaly field is multiplied by unknown (i.e., adjustable by the inversion) scaling factors $\gamma_{\mathrm{NEE-T}}$ (the NEE-T regression coefficients). These scaling factors are identical in each year of the inversion, but are allowed to vary smoothly both seasonally (with a correlation length of about 3 weeks, such that $\gamma_{\mathrm{NEE-T}}$ contains 13 independent degrees of freedom in time, repeated every year) and spatially (with correlation lengths of about $1600\,\mathrm{km}$ in longitude direction and $800\,\mathrm{km}$ in latitude direction, imposing a spatial smoothing on $\gamma_{\mathrm{NEE-T}}$ over the same spatial scales as the smoothing imposed on the interannual flux anomalies $\mathbf{f}_{\mathrm{NEE,IAV}}^{\mathrm{adj}}$ in the standard inversion). The need for seasonal and spatial resolution of $\gamma_{\mathrm{NEE-T}}$ has been inferred from analysis of the standard inversion results (Sect. 2.1). The a-priori spatial and temporal correlations are imposed on $\gamma_{\mathrm{NEE-T}}$ to prevent a localization of inverse adjustments in the vicinity of the atmosperic stations. In contrast to the standard inversion, however, where the a-priori correlations lead to a smooth NEE field, the NEE result of the NEE-T inversion still retains structure on the pixel and monthly scale from the temperature field. By having only 13 degrees of freedom in the time dimension, the introduction of the regression term also regularizes the inversion further compared with the explicit interannual term of the standard inversion, which has 796 degrees of freedom in the time dimension.

Eq. (2) also contains adjustable residual terms (2nd line) to accomodate modes of variability from the atmospheric $CO_2$ signals that cannot be explicitly represented by the regression term and might therefore be at risk of being aliased into spurious adjustments to $\gamma_{\mathrm{NEE-T}}$:

– Outside the non-zero period 1985-2016 of the regression term, interannual NEE variations are represented by a standard interannual term $\mathbf{f}_{\mathrm{NEE,IAV}}^{\mathrm{adj}}$ with weights $(1-w)$ opposite to those of the regression term.

– An adjustable linear trend ($\mathbf{f}_{\mathrm{NEE,Trend}}^{\mathrm{adj}}$) is needed because trends have explicitly been removed from $\mathbf{T}$. For every pixel, $\mathbf{f}_{\mathrm{NEE,Trend}}^{\mathrm{adj}}$ is proportional to the time difference $\Delta t$ since the beginning of the calculation period, multiplied by an unknown trend parameter to be adjusted by the inversion (with zero prior). The trend parameters are correlated with each other in space with the same correlation length scale as the mean and interannual variability components of the standard inversion (i.e., as $\mathbf{f}_{\mathrm{NEE,LT}}^{\mathrm{adj}}$ and $\mathbf{f}_{\mathrm{NEE,IAV}}^{\mathrm{adj}}$ in Eq. (1)).

– Further, as the NEE field from the standard inversion contains a strong increase in seasonal cycle amplitude in northern extratropical latitudes (earlier described in Graven et al. (2013); Welp et al. (2016)) which is expected to not (solely) arise from changes in the temperature seasonal cycle, we decoupled this mode of variability from the regression by adding it as an explicitly adjustable term $\mathbf{f}_{\mathrm{NEE,SCTrend}}^{\mathrm{adj}}$. For each degree of freedom in the mean seasonality term $\mathbf{f}_{\mathrm{NEE,Seas}}^{\mathrm{adj}}$ in Eq. (1)), the additional term $\mathbf{f}_{\mathrm{NEE,SCTrend}}^{\mathrm{adj}}$ contains the same mode multiplied by $\Delta t$ and having its own adjustable strength parameter.

Any further residual modes of variability (including NEE variations related to variations in other environmental drivers uncorrelated to $\mathbf{T}$ variations, non-linear responses, memory effects and internal ecosystem dynamics, errors in the employed $\mathbf{T}$ field, errors of the a-priori fixed ocean and fossil fuel terms, as well as effects of transport model errors) are not explicitly accounted for, as we lack sufficient a-priori information to model them explicitly. To the extend that they are uncorrelated to $\mathbf{T}$ variations, they will stay in the data residual of the inversion.

In contrast to the standard inversion using 23 stations with temporally homogeneous records over 1985-2016, the NEE-T inversion uses atmospheric data from 89 stations (Table 1) partially with shorter records but spatially covering the globe more evenly (including stations in northern Siberia and tropical America). While the standard inversion with explicitly time-dependent degrees of freedom can develop spurious NEE variations when stations pop in or out with time, the major interannual variability from the NEE-T inversion is coming from the regression term using its degrees of freedom ($\gamma_{\mathrm{NEE-T}}$) repeatedly each year, such that any data point influences all years of the calculation period simultaneously. Therefore, the NEE-T inversion is not prone to spurious variations from a temporally changing station network.

## 2.3 Sensitivity cases

The algorithm uses several inputs carrying uncertainties, and contains several parameters that are not well determined from a-priori available information. Therefore, we also ran an ensemble of sensitivity cases. In each such sensitivity case, one of the uncertain elements of the algorithm is changed within ranges that may be considered as plausible as the base case: (1) longer spatial a-priori correlations (2.4 times in longitude direction and 1.6 times in latitude direction) for $\gamma_{\mathrm{NEE-T}}$, (2) 4 weeks (rather than 3 weeks) temporal a-priori correlation length scale for $\gamma_{\mathrm{NEE-T}}$, (3) halved a-priori uncertainty range for $\gamma_{\mathrm{NEE-T}}$, (4) using ocean $CO_2$ fluxes from the PlankTOM5 ocean biogeochemical process model (Buitenhuis et al., 2010) instead of the fluxes based on $pCO_2$ measurements, (5) taking the gridded monthly land temperature field from Berkeley Earth

(www.BerkeleyEarth.org, accessed 2017-11-29) instead of the GISS data set, and (6) using ERA-Interim meteorological fields (Dee et al., 2011) to drive the atmospheric transport model rather than NCEP meteorological fields.

Eight additional sensitivity cases have been run to demonstrate coherent information in the atmospheric data. The set of 89 stations used in the base case was divided into 8 mutually exclusive parts (Table 1). In each of the sensitivity cases, one of these parts was omitted, leaving sets of 73 to 82 remaining stations. By this construction, all these 8 runs still have global data coverage, but every station is absent in one of the runs. If the results would depend on any particular station without being backed up by other stations, then the run omitting this station would show substantial difference from the base run.

The range of results from this ensemble of sensitivity cases will be shown as uncertainty range around the base case.

## 2.4   Comparison to eddy covariance data

For comparison of the estimated sensitivities $\gamma_{\text{NEE-T}}$ against independent information, we also calculate NEE-T relationships from eddy covariance (EC) measurements. We use NEE and co-measured air temperature records from the FLUXNET2015 data set (https://fluxnet.fluxdata.org). EC sites (Table 2) have been chosen based on having long records (at least 12 years; 2 sites with 11 years were included too to have more ecosystem types represented). Crop sites have not been included because their flux variability may strongly depend on crop rotation.

We start from the half-hourly or hourly data sets (variables NEE_CUT_REF and TA_F_MDS, respectively). Records classified as "measured" (QC flag = 0) or "good quality gapfill" (QC flag = 1) in both variables are averaged over each month. Months with data coverage of 90% or less are discarded from the statistical analysis.

For each EC site and each month of the year, all available monthly $CO_2$ flux values from the different years were regressed against the corresponding monthly air temperature values, using ordinary least squares regression. This yields sensitivities as regression slopes $g_{\text{NEE-T}}^{\text{EC}} = \Delta\text{NEE}^{\text{EC}} / \Delta\text{T}^{\text{EC}}$. We also calculated the confidence interval of the slope for the confidence level 90%, reflecting the uncertainty of $g_{\text{NEE-T}}^{\text{EC}}$ given the scatter of the monthly values around a linear relationship.

The sensitivities $\gamma_{\text{NEE-T}}$ from the inversion and $g_{\text{NEE-T}}^{\text{EC}}$ from the explicit linear regression are not fully comparable mathematically because (i) the time period (and to some extent the frequency filtering) are different, and (ii) the explicit linear regression of the total NEE is not only influenced by the year-to-year variations but also by the ratio of NEE trend and temperature trend while $\gamma_{\text{NEE-T}}$ has deliberately been made insensitive to the trend (Sect. 2.2). Therefore, we also calculated sensitivities $g_{\text{NEE-T}}^{\text{Inv}}$ from the total monthly-mean non-fossil $CO_2$ flux (i.e., including regression and residual

terms of Eq. (2)) and the employed temperature field of the inversions, in the same way and subsampled at the same months as for the EC data. A perfect match between $g_{\text{NEE-T}}^{\text{EC}}$ and $g_{\text{NEE-T}}^{\text{Inv}}$ cannot be expected nevertheless because (iii) sensitivities from the inversion even at its smallest resolved scale –the pixel scale– represent a mixture of ecosystem types in unknown proportions, while the EC data represent a specific ecosystem type, (iv) NEE from the inversion includes the effects of disturbances such as fire, which are absent from the EC data, and (v) there may be local trends in the ecosystem behaviour observed by the EC data due to aging or slow species shifts, which average out on the larger spatial scales seen by the atmospheric inversion.

## 3   Results

### 3.1   How does the "interannual climate sensitivity" $\gamma_{\text{NEE-T}}$ vary in space and by season?

As a starting point, we present the results of the NEE-T inversion in terms of $\gamma_{\text{NEE-T}}$, which is the local regression coefficient between interannual variations in NEE and temperature, resolved seasonally (Sect. 2.2). As $\gamma_{\text{NEE-T}}$ does not only reflect direct temperature responses but also responses to other environmental variables that co-vary with temperature (such as water availability, incoming solar radiation), we refer to it as "interannual climate sensitivity".

Fig. 1 presents the seasonal and spatial patterns of the "interannual climate sensitivity" as Hovmöller Diagrams, showing longitudinally averaged $\gamma_{\text{NEE-T}}$ in dependence on latitude and month-of-year. The longitudinal average is taken separately over North and South America (left panel), Europe and Africa (middle panel), and Asia and Australia (right panel), respectively. This representation summarizes the essential variations of $\gamma_{\text{NEE-T}}$, as it is found to be relatively uniform across longitude within the individual continents (not shown).

In essentially all *northern extratropical land* areas (north of about 35° N), we estimate negative $\gamma_{\text{NEE-T}}$ in spring (and, to a lesser extent, autumn), consistent with photosynthesis being temperature limited such that higher-than-normal temperatures lead to more negative NEE (i.e., larger-than-normal $CO_2$ uptake) and vice versa. Warmer conditions tend to coincide with higher incoming solar radiation in May and/or June in the northern extratropics (according to a correlation analysis of CRUNCEPv7 data, not shown), which would tend to amplify the direct temperature effect. In summer, when photosynthesis is not limited by temperature any more, we find positive $\gamma_{\text{NEE-T}}$ values. Such positive $\gamma_{\text{NEE-T}}$ is consistent with enhanced respiration in warmer summers, but also with the fact that warmer-than-normal periods are often also dryer leading to reduced photosynthetic uptake or enhanced fire activity. In winter, NEE is not found to respond much to interannual climate variations. The interpretation of the sea-

sonality of $\gamma_{NEE-T}$ is confirmed by its latitude dependence: Consistent with the later spring and shorter summer in the higher northern latitudes, the period of negative $\gamma_{NEE-T}$ starts later there, and the period of positive $\gamma_{NEE-T}$ is shorter.

In the *Tropics*, we find stronger and less systematic variations in $\gamma_{NEE-T}$. However, as indicated by the missing stippling, we also find larger disagreement between our sensitivity cases designed to embrace plausible ranges for the essential inputs and parameters in the algorithm (Sect. 2.3). This reveals that the seasonal variations in $\gamma_{NEE-T}$ are of limited robustness here. Nevertheless, a clear feature in the tropics is the dominance of positive $\gamma_{NEE-T}$ values.

In *extratropical South America and Africa*, the seasonal pattern has similarities with the northern extratropical pattern shifted by 6 months. The pattern in *Australia* is difficult to interpret, but also not very robust. Larger errors in the southern extratropics may concievably arise because the much smaller land area involves a much smaller number of degrees of freedom available to satisfy the data constraints (remember that the oceanic flux cannot be adjusted in this inversion, while the $pCO_2$-based ocean prior flux is actually less well constrained in the southern extratropics due to the much smaller density of $pCO_2$ data).

### 3.2 How much interannual variability of NEE can be reproduced by the seasonally resolved linear regression to T?

The assumed linear relationship between NEE anomalies and air temperature anomalies around their respective seasonal cycles represents a strong abstraction of the complex underlying physiological and ecosystem processes. Nevertheless, the interannual variations of global total NEE estimated by the NEE-T inversion is very similar to that estimated by the standard inversion (Fig. 2, top left). The agreement is confirmed by high correlation (Fig. 2, top right). For interpretation, we note that variations in the global total $CO_2$ flux are very well constrained from atmospheric $CO_2$ observations at time scales longer than the atmospheric mixing time (about 4 years) (Ballantyne et al., 2012). Variations on the year-to-year scale are tightly constrained already (Peylin et al., 2013). We thus use the global $CO_2$ flux from the standard inversion having explicit interannual degrees of freedom as a benchmark. Since the ocean flux is identical in both standard and NEE-T inversion runs, the high level of agreement in Fig. 2 (top) means that the spatially and seasonally resolved linear NEE-T regression provides already a good approximation to global interannual NEE variations.

Almost the same level of agreement is also found for a split of the global NEE into a northern extratropical and a tropical plus southern extratropical contribution (Fig. 2, middle and bottom). Due to the faster atmospheric mixing within the extratropical hemispheres compared to the mixing across latitudes, these two NEE contributions are expected to be relatively well constrained by atmospheric data independently of

each other. The linear approximation of the NEE-T inversion is able to distinguish extratropical and tropical behaviour.

For a further split into smaller regions, in particular along longitude, interannual NEE variations from standard and NEE-T inversions stay similar, but deviations get larger (not shown). This could indicate that the limits of the linear NEE-T relationship start to kick in at these scales. However, the NEE variations cannot be expected to be well constrained from the atmospheric data at the regional scale any more. Thus, the discrepancy can also be caused by the standard inversion, while the NEE-T inversion could be the more realistic one by profiting from the pixel-scale information added through the temperature field, as discussed in Sect. 4.1.

### 3.3 Are the estimated patterns of $\gamma_{NEE-T}$ compatible with ecosystem-scale eddy covariance data?

Fig. 3 compares "interannual climate sensitivities" (ordinate) calculated by the NEE-T inversion with those calculated independently from eddy covariance (EC) data for each month of the year (abscissa). Each panel represents an EC site, roughly arranged by ecosystem types and latitudes. The orange line with the surrounding gray band give the sensitivities $\gamma_{NEE-T}$ from the various NEE-T inversion runs as in Fig. 2 taken at the respective pixels enclosing the EC sites. The black dots are the sensitivities $g_{NEE-T}^{EC}$ calculated by explicit linear regression of monthly EC flux records against the co-measured monthly air temperature (Sect. 2.4).

To allow a fairer comparison between inversion results and EC data, additional color dots give sensitivities $g_{NEE-T}^{Inv}$ calculated from the NEE-T inversion results in the same way and subsampled at the same months as for the EC data (Sect. 2.4). At most EC sites, the sensitivities calculated by the inversion itself ($\gamma_{NEE-T}$, orange lines) or by explicit regression afterwards ($g_{NEE-T}^{Inv}$, orange dots) mostly agree within the confidence interval of the regression. This shows that the comparison of inversion and EC sensitivities is meaningful despite their differences in meaning and calculation (in particular, the trend influence (issue (ii) in Sect. 2.4) on $g_{NEE-T}^{Inv}$ turns out to be relatively small because the explicit regressions are only done over the limited time period spanned by the EC records).

Despite their completely independent sources of information and their remaining incompatibilities (Sect. 2.4), the sensitivities from the EC data and the atmospheric NEE-T inversion have a similar order of magnitude as well as similar seasonal patterns for a majority of EC sites (Fig. 3). For most sites/months, the sensitivities agree within their confidence intervals. The level of agreement roughly depends on ecosystem type and latitude:

- Generally good consistency is found in high northern latitudes (line 1 of panels in Fig. 3) and at evergreen needleleaf forest (ENF) sites in temperate northern latitudes (line 2 and rightmost part of line 3).

– At mixed forest (MF) and decidious broadleaf forest (DBF) sites in temperate northern latitudes (left part of line 3 and line 4), consistency is mostly good as well, though some months in spring or summer have more negative $g_{\text{NEE-T}}^{\text{Inv}}$ sensitivities from EC data (e.g., DE-Hai, DK-Sor, BE-Bra). However, the behaviour of DBF ecosystems is not an important contribution to larger-scale NEE variability because DBF ecosystems only cover $11\%$ to $25\%$ of the area around the sites shown.

– Generally good consistency within the confidence interval is also found at sites of various other ecosystem types in temperate northern latitudes (line 5).

– At the tropical and southern extratropical sites (last line), the comparison does not yield conclusive information, because the confidence intervals of the regression are much larger than the seasonal variations of both inversion and EC results. We can only state that the $g_{\text{NEE-T}}^{\text{Inv}}$ and $g_{\text{NEE-T}}^{\text{EC}}$ sensitivities do not contradict each other statistically. Some qualitative consistency is found at the Australian EBF site, even though the dominant vegetation round the site is shrubland (about $45\%$).

Though this comparison partly remains inconclusive (as the confidence intervals at tropical and southern hemispheric sites are large, as $g_{\text{NEE-T}}^{\text{Inv}}$ and $g_{\text{NEE-T}}^{\text{EC}}$ are not actually fully comparable (Sect. 2.4), and as by far not all areas and dominating ecosystem types are represented), it does support the results of the NEE-T inversion at least in the northern extratropics.

## 4 Discussion

### 4.1 NEE variations in the northern extratropics

Given that we found robust seasonal patterns of $\gamma_{\text{NEE-T}}$ which can be interpreted in terms of the fundamental physiological processes (Sect. 3.1), that these patterns are compatible with inferences from independent ecosystem-scale eddy covariance (EC) measurements (Sect. 3.3), and that the corresponding interannual NEE variations are compatible with the atmospheric constraint on the most reliable large scales (Sect. 3.2), we conclude that the linear dependence of NEE anomalies on air temperature anomalies (as climate proxy) represents a meaningful approximative empirical description of the northern extratropical biosphere. The compatibility of the NEE-T relationships inferred from large-scale atmospheric constraints and ecosystem-scale EC constraints of dominating vegetation types suggests that the regional or continental NEE variations are to a substantial degree due to local variations linked to local climate anomalies; otherwise the NEE-T inversion could not have worked. Given that, we expect the NEE-T inversion to provide more realistic interannual NEE variations on regional scales than the standard inversion which smoothly interpolates NEE on scales smaller

than station-to-station differences (compare last paragraph of Sect. 3.2).

Note that, as EC data measure fluxes on small spatial scales (a few 100 meters), the EC flux variations themselves cannot directly be compared to the inversion results representing NEE over (sub)continental scales and integrating over many ecosystem types and climate regimes. In contrast to the fluxes, however, derived relationships (such as the NEE-T relationships considered here) may well be able to bridge this scale gap.

Besides the interannual variations, the NEE-T inversion also reproduces the small negative trend in NEE through its residual term $f_{\text{NEE,Trend}}^{\text{adj}}$ in Eq. (2) (Fig. 2). Likewise, it reproduces the northern extratropical increase in seasonal cycle amplitude through its residual term $f_{\text{NEE,SCTrend}}^{\text{adj}}$ (not shown).

### 4.2 NEE variations in the tropics

In contrast to the northern extratropics, we did not find conclusive seasonal patterns of $\gamma_{\text{NEE-T}}$ in the tropics (Sect. 3.1). However, despite the substantial uncertainty range of $\gamma_{\text{NEE-T}}$ (Fig. 1), the sensitivity cases reproduce almost identical interannual NEE variations in the tropics (see the narrow gray band round the NEE-T estimate in Fig. 2, bottom left). This underlines that pan-tropical NEE variations are actually well constrained from the atmospheric data, while the seasonal differences in $\gamma_{\text{NEE-T}}$ arise to compensate for the set-up differences among the sensitivity cases. As shown below (Sect. 4.3), all the seasonally different $\gamma_{\text{NEE-T}}$ estimates correspond to a similar effective sensitivity (having a positive value) on yearly time scales. Due to this, the NEE-T inversion is found to possess predictive skill on the time scale of El Niño / Southern Oscillation (Rödenbeck et al., 2018).

The positive effective $\gamma_{\text{NEE-T}}$ in the tropics (Sect. 3.1) is consistent with the strong positive correlation of atmospheric $CO_2$ growth with large-scale tropical annual temperature (Wang et al., 2013). This is unlikely to arise from a direct temperature effect, however, because process studies (e.g., Meir et al., 2008; Bonal et al., 2008; Alden et al., 2016) point to water availability, rather than temperature, as the dominant control on the ecosystem scale. This is also confirmed by the large confidence intervals of the NEE-T regression of the EC data from the only tropical site available here (GF-Guy, leftmost on last line of Fig. 3). A strong correlation to temperature can still arise statistically due to the strong link of temperature and precipitation anomalies over larger spatial scales (Berg et al., 2014). Moreover, vapour pressure deficit (VPD) controlling photosynthesis responds to temperature variations particularly strongly in the warm tropical climate due to the non-linearity of the VPD(T) dependence (Monteith and Unsworth, 1990). Further, T is spatially coherent over much larger areas in the tropics while variability in water availability is local and averges out over larger spatial scales (Jung et al., 2017). Nevertheless, even a direct temperature effect in the tropics was found by Clark et al.

(2013) at least for a component flux of NEE (wood production) in 12-year plot data.

## 4.3 An extended benchmark for process models

Empirical data-based relationships between interannual NEE variations and air temperature variations have been proposed in the literature as benchmarks to evaluate biogeochemical process models. For example, Cox et al. (2013) calculated an effective global climate sensitivity of $5.1\pm0.9\,\mathrm{PgC\,yr^{-1}K^{-1}}$ over 1960-2010, regressing the annual $CO_2$ growth rate observed at the station Mauna Loa (Hawaii) (taken as a proxy for the global total $CO_2$ flux) against $30°\,\mathrm{N}$–$30°\,\mathrm{S}$ (both land and ocean) averaged air temperature (after detrending both time series by subtracting an 11-year running mean). In a similar way (using the average atmospheric growth rate from a varying set of background sites, a slightly different time series treatment, and $24°\,\mathrm{N}$–$24°\,\mathrm{S}$ land temperature), Wang et al. (2013) obtained a value of $3.5\pm0.6\,\mathrm{PgC\,yr^{-1}K^{-1}}$ over 1959-2011. Wang et al. (2014) regressed the mean Mauna Loa and South Pole $CO_2$ growth rates against $23°\,\mathrm{N}$–$23°\,\mathrm{S}$ vegetated land temperature over moving 20-year windows and reported effective global climate sensitivities between $3.4\pm0.4\,\mathrm{PgC\,yr^{-1}K^{-1}}$ during 1960-1979 and $5.4\pm0.4\,\mathrm{PgC\,yr^{-1}K^{-1}}$ during 1992-2011.

The inversion results presented here allow to extend these benchmarks in two ways. As a first extension, we can evaluate to which extent the interannual variations in local or averaged atmospheric $CO_2$ growth rates are indeed equivalent to the interannual variations in the global total $CO_2$ flux (as implicitly assumed in the above-mentioned studies), and to which extent the global total $CO_2$ flux is indeed representative for global terrestrial NEE or, even more specifically, for tropical NEE. This can be evaluated here because all these time series (spatially explicit $CO_2$ fluxes with all their contributions, as well as the corresponding atmospheric $CO_2$ variations at the measurement stations) are available within the inversion calculation. To ensure a mutually consistent treatment of these time series, we used running yearly averages (January-through-December, February-through-next-January, etc.) of the flux time series and running yearly differences (next-January-minus-January, next-February-minus-February, etc., multiplied by $2.12\,\mathrm{PgC\,ppm^{-1}}$ (Ballantyne et al., 2012)) of the atmospheric $CO_2$ time series, respectively. All these interannual time series were then regressed over 1985-2016 against annual tropical land temperature ($25°\,\mathrm{N}$–$25°\,\mathrm{S}$) derived from the same temperature field without decadal variations as used in the NEE-T inversion. The resulting effective climate sensitivities are shown in Fig. 4. The sensitivities of the total $CO_2$ flux (solid bars in the middle) calculated from the standard inversion (black) or from the NEE-T inversion (orange) are similar to each other, and fall in between the values by Cox et al. (2013) and Wang et al. (2013). Part of the discrepancies between these results can be attributed to the different

time periods and the different time series treatments (in particular, to the extent to which decadal variability has been removed). Fig. 4, however, reveals another reason of the discrepancies: The sensitivity of the Mauna Loa growth rate (middle one of the hashed blue bars) is larger than that of the global flux (solid bars). This cannot be due to a deficiency of the inversions to fit Mauna Loa's variability, because the modelled Mauna Loa sensitivities (hashed bars next to the middle blue bar) agree well with the observed one. Thus, a sensitivity calculated from the Mauna Loa growth rate (as in Cox et al., 2013) somewhat overestimates the sensitivity of the global flux. The Mauna Loa sensitivity is still much closer to that of the global $CO_2$ flux than sensitivities calculated from most other stations: Southern extratropical stations like South Pole (or from the Mauna Loa and South Pole mean as in Wang et al., 2014) lead to a substantial underestimation (it is unclear why the sensitivity reported by Wang et al. (2014) for the recent 1992-2011 period is nevertheless even higher than our Mauna Loa value), while northern extratropical stations like Point Barrow lead to an even stronger overestimation than Mauna Loa. This suggests that using a varying mixture of stations (as in Wang et al., 2013) can induce further errors, in particular when possible changes in sensitivity are considered. We note that the atmospheric inversions benefit from using multiple station records, because the transport model links the atmospheric $CO_2$ signals to their different areas of origin, rather than the instantaneous link of the atmospheric signals to the global flux as in the direct use of station records.

Care is also needed in the interpretation of the estimated effective sensitivities: The sensitivity of the total $CO_2$ flux (solid bars) underestimates that of global NEE only (horizontally hashed bars), because the ocean flux is substantially anti-correlated to NEE on the interannual time scale. The sensitivity of tropical-only NEE (vertically hashed bars) is smaller than that of global NEE, though the reduction is less than according to the ratio of land area, confirming the dominance of tropical NEE variations.

As a second extension of process model benchmarking, the data-based estimates of the spatially and seasonally resolved $\gamma_{\mathrm{NEE\text{-}T}}$ from the NEE-T inversion can directly be employed as target values, by regressing the NEE simulated by the terrestrial biosphere or Earth system model against the model temperature for individual small regions and seasons across the years 1985-2016 and comparing these model-derived local and season-specific sensitivities to the data-based values presented here (using the ensemble of sensitivity cases as a measure of uncertainty in $\gamma_{\mathrm{NEE\text{-}T}}$). Importantly, before regressing, the model NEE and temperature fields need to be deseasonalized, detrended, and filtered in the same way as done for the observed temperature in the NEE-T inversion (Sect. 2.2), because the numerical $\gamma_{\mathrm{NEE\text{-}T}}$ values are somewhat specific to the chosen filtering, in particular to the exact way to remove decadal variations (as is the case also for the effective global climate sensitivity tar-

gets by Cox et al. (2013), Wang et al. (2013), and Wang et al. (2014)). For the northern extratropics, where $\gamma_{\text{NEE-T}}$ is quite robustly constrained and shows distinct spatial and seasonal patterns (Sect. 3.1), this offers a much more detailed benchmark of the process representation in the models than the existing single-valued effective climate sensitivity of the global $CO_2$ growth rate. For the tropics, unfortunately, $\gamma_{\text{NEE-T}}$ is not constrained well enough to do that, but due to the fact that pan-tropical NEE variations are nevertheless quite robust (Sect. 4.2), the effective climate sensitivity of tropical NEE from Fig. 4 ($4.2\,\text{PgC}\,\text{yr}^{-1}\text{K}^{-1}$ with a range across the sensitivity cases of $3.8\ldots4.4\,\text{PgC}\,\text{yr}^{-1}\text{K}^{-1}$) may be used as a specifically tropical target instead.

### 4.4 Could the results be improved by using a multivariate regression against further climatic variables?

We tested the algorithm also with precipitation (P) or solar radiation as explanatory variables, individually or in multivariate combinations (not shown). While, for example, an NEE-P inversion had almost as good an explanatory power as the NEE-T inversion, a multivariate NEE-T-P inversion did not explain much more NEE variations than the univariate NEE-T inversion did already. This confirms the strong background correlations of air temperature with the other climate variables on interannual time scales. It also means that a multivariate regression would –despite a mathematically unique partitioning into contributions of the individual explanatory variables– likely not yield an uniquely interpretable attribution of NEE variability to different causes.

Given that, a univariate NEE-T inversion seems advantageous because T likely has data sets best constrained by observations. As a regression is confined to the variability present in the explanatory variables, using less well observed or even modelled variables (as would be the case for precipitation or cloud cover) involves the risk of contamination.

## 5 Conclusions and outlook

The response of Net Ecosystem Exchange (NEE) to climate anomalies has been estimated by linear regression against anomalies in air temperature (T) within an atmospheric inversion based on a set of long-term atmospheric $CO_2$ observations. The resulting spatially and seasonally resolved regression coefficients $\gamma_{\text{NEE-T}}$ are interpreted as a "interannual climate sensitivity", comprising the direct temperature response as well as responses to covarying anomalies in other environmental conditions (e.g., moisture, radiation) (Sect. 4.4).

– The inferred "interannual climate sensitivity" $\gamma_{\text{NEE-T}}$ shows distinct and interpretable patterns along latitude and season. In particular, we find negative $\gamma_{\text{NEE-T}}$ during spring and autumn (consistent with a temperature-limited photosynthesis) and positive $\gamma_{\text{NEE-T}}$ during summer (consistent with a water-limited photosynthesis) in all northern extratropical ecosystems (Sect. 3.1).

– Despite the complexity of the underlying plant and ecosystem processes, the spatially and seasonally resolved linear regression of NEE against temperature anomalies (taken as climate proxy), fitted to atmospheric $CO_2$ data, can reproduce a large fraction of NEE's interannual variations, at least in the northern extratropics. This conclusion is based on the agreement of the inferred NEE variations with a time-explicit atmospheric inversion at well-constraint large spatial scales (Sect. 3.2), and the consistency of $\gamma_{\text{NEE-T}}$ with independent calculations from eddy covariance data at small spatial scales (Sect. 3.3). Among the reasons for this potentially surprising finding is that the regression is only applied to the interannual anomalies of NEE around its mean seasonal cycle (rather than to the full range of seasonal temperature variations), and that the different behaviours in different seasons have been accounted for.

The results of the NEE-T inversion can be applied to benchmark process models of the land biosphere or Earth system models: The spatially and seasonally resolved interannual climate sensitivity $\gamma_{\text{NEE-T}}$ can be calculated from the model output (using detrended NEE over the period 1985-2016 for consistency) and compared to the values presented here; this allows a more detailed benchmark for the northern extratropical ecosystem processes than existing effective global sensitivities. Further, as its adjustable degrees of freedom are identically applied every year, the regression offers a way to bridge temporal gaps in the atmospheric $CO_2$ records; it transfers information from the recent data-rich years into the more data-sparse past. Similarly, the NEE-T regression allows to forecast the $CO_2$ flux for some years, if forecasted air temperatures (and extrapolations of fossil fuel emissions and the ocean exchange) are available. As another application, the regression may help to uncover smaller decadal trends in the atmospheric $CO_2$ signal by separating them from the larger interannual responses of NEE. Extending the calculation to the full period of atmospheric $CO_2$ measurements (since the late 1950ies, see Rödenbeck et al. (2018)), we can investigate possible decadal changes in the interannual climate sensitivity $\gamma_{\text{NEE-T}}$.

The inversion results are available for use in collaborative projects from the Jena CarboScope website http://www.BGC-Jena.mpg.de/CarboScope/.

## Appendix A: More specification details of the inversion algorithm

This appendix first reviews the base set-up and implementation of the Jena CarboScope atmospheric $CO_2$ inversion in its current version v4.1, from which the particular runs used in this study are derived (Sect. A1). Sect. A2 gives

differences of the run s85oc_v4.1s used as standard inversion here. The further differences of the NEE-T inversion s85ocNEET_v4.1s have already been described in Sect. 2.2.

For more details, formulas, or deeper explanations, the reader is referred to the technical report Rödenbeck (2005).

## A1 The Jena CarboScope atmospheric $CO_2$ inversion v4.1

The Jena CarboScope $CO_2$ inversion is a linear Bayesian atmospheric inversion, estimating land-atmosphere and ocean-atmosphere $CO_2$ fluxes from long-term atmospheric $CO_2$ mole fraction measurements (Rödenbeck, 2005). As the Jena CarboScope is particularly focused on interannual variations, flux estimates are only used over time periods homogeneously covered by all data records, to avoid spurious jumps (or changes in the amplitude of variations) that can result from changes of the station set over time. To deal with the fact that many of today's measurement stations came into operation at various points in time during the last decades, the Jena CarboScope provides several runs, either over longer periods (the longest one currently being 1976-2016) with only a few stations, or runs with more stations (currently up to 59) but correspondingly shorter periods. Despite these different "periods of validity", however, all base runs are carried out over 1955-2017, which includes time for spin-up and spin-down to minimize "edge effects". The Jena CarboScope inversion is regularly updated, mostly yearly to include the latest year of measurements. These updates may also involve some changes in the station sets according to data availability, as well as changes in the inversion set-up and implementation details. All results are available for use in collaborative projects from the Jena CarboScope website http://www.BGC-Jena.mpg.de/CarboScope/.

The following provides some specification details as of the current version v4.1 of the CarboScope inversion, also pointing out changes with respect to the previous version v3.8.

### A1.1 Grid resolution

The $CO_2$ fluxes have a daily time resolution and are represented on the grid of the transport model ($\approx 4° \times 5°$, see below).

### A1.2 Prior information

Bayesian prior information is used to regularize the otherwise underdetermined estimation. However, none of the basic CarboScope inversion runs involves any information from terrestrial and oceanic carbon cycle models, in order to transparently base the results on atmospheric information and thus to allow independent comparison to process models or to empirical models like the NEE-T inversion.

The a-priori probability distribution of the fluxes is not directly implemented through a covariance matrix, but indirectly through a statistical "flux model" that expresses the spatio-temporal $CO_2$ flux field as a linear function of a vector of independent adjustable dimensionless parameters with zero mean and unit variance. This makes it easy to specify, e.g., time-scale dependent statistical properties, or to simultaneously specify temporal and spatial a-priori correlations.

The prior flux of all *land NEE components* is zero. This means that the "error" of this prior is identical with the land $CO_2$ flux itself, i.e., the a-priori probability density describes expected statistical properties of NEE. Its a-priori uncertainties are proportional to the fraction of vegetated land area in each pixel, taken as the sum of 'crop', 'dbf', 'dnf', 'ebf', 'enf', 'grass', and 'shrub' fractions from SYNMAP (Jung et al., 2006). The results of the v4.1 inversions on larger spatial scales are still quite similar to version v3.8 (which still used spatial patterns of a-priori uncertainty derived from model output), confirming that the variability was not driven by these spatial patterns. The largest difference of v4.1 results to previous versions is a smaller amplitude of interannual variations in the tropical land fluxes.

NEE adjustments are split into the temporal mean, a large-scale mean seasonality, and (interannual) variations. The large-scale mean seasonality has a-priori correlations of about $3825\,\mathrm{km}$ longitudinally, $1275\,\mathrm{km}$ latitudinally, and about 4 weeks in time. The correlation lengths of the other two flux contributions are about $1600\,\mathrm{km}$ longitudinally and about $800\,\mathrm{km}$ latitudinally; and in the "variations" part 2 weeks in time. For practical reasons, the temporal variations in all adjustable terms are implemented as Fourier series. The temporal correlations can then simply be implemented by downweighting the a-priori uncertainties of the Fourier modes with higher frequencies according to the spectrum corresponding to the desired autocorrelation function. The split into long-term, seasonal, and non-seasonal contributions can be implemented just by only activating the corresponding part of the Fourier series. Note that not only the "mean seasonality" part but also the "variations" part contains seasonal Fourier terms, to allow seasonal variability also to be adjusted on the smaller spatial scales.

*Ocean fluxes* are implemented analoguosly to land NEE, with a-priori uncertainties proportional to the ocean fraction, and slightly longer a-priori spatial correlations (about $1912\,\mathrm{km}$ longitudinally and about $956\,\mathrm{km}$ latitudinally). In contrast to land NEE, however, the mean spatial flux pattern and its mean seasonal cycle are not adjusted, but prescribed to the mean seasonal cycle of the flux estimates oc_v1.4 (update of Rödenbeck et al., 2014) based on an interpolation of $pCO_2$ data from the SOCATv4 data base (Bakker et al., 2016). Only the (interannual) ocean flux variability can be adjusted by the inversion in the basic v4.1 runs (see the difference in the present "standard inverion" in Sect. A2 below).

The *fossil fuel emission* prior is taken from monthly values of CDIAC (Andres et al., 2016). The years after 2013 have been extrapolated by global scaling factors based on the ratios in the emission totals from Le Quéré et al. (2016, update

for year 2016). There are no inverse adjustments to fossil fuel emissions.

### A1.3    Data treatment

The CarboScope inversion uses the individual data points in the atmospheric $CO_2$ records (flask pair values or hourly averages, respectively). In order to avoid that the in-situ records with hourly data dominate the result, a "data density weighting" has been implemented. It artificially increases the model-data mismatch uncertainty of data points from dense records in such a way that weekly periods of data always have the same impact on the results.

The individual $CO_2$ data points are *screened for outliers* by a "$2\sigma$ criterion" (newly introduced in CarboScope version v4.1): A pre-run of the inversion is done, using the base CarboScope set-up and a large set of stations potentially used in later runs. Then, the $CO_2$ mole fraction residuals between a forward run from the posterior fluxes and the data are considered. For each station, data points are removed if their residual is larger than 2 standard deviations across all residuals of that station. This procedure is similar to the outlier flagging done routinely by many atmospheric data providers. By doing it within the inversion, the deficiencies of the transport model to reproduce small-scale circulation are taken into account to some extent. The procedure can also be understood as an approximate way to implement a non-Gaussian probability density for the model-data mismatch: As residuals larger than $2\sigma$ are very unlikely in the Gaussian distribution, an inversion assuming Gaussian model-data mismatches will respond strongly to "outliers" to reduce these mismatches; in contrast, the "$2\sigma$ screening" effectively assigns an infinitely large uncertainty to these data points. The results mostly stay similar after this screening, but some flux anomalies get removed. In most cases, these anomalies were unrobust, in that they were dampened much faster than other anomalies when increasing the strength of the prior constraint (parameter $\mu$ in Rödenbeck (2005)). For example, many of the spikes in the $CO_2$ record of station KEY and their effect on the $CO_2$ flux estimates for northern temperate America are removed by the screening. We interpret these spikes as influence of local fossil fuel emissions, which would be mistaken by the inverison as regional signals. This interpretation is supported by the fact that more and more of these spikes occur in the more recent decades. The introduction of the $2\sigma$ screening made it possible to re-add further stations with pronounced spikes, such as station TAP.

### A1.4    Further implementation details

*Atmospheric tracer transport* in the global CarboScope inversions is simulated by the TM3 model (Heimann and Körner, 2003) (resolution $\approx$ 4° $\times$ 5° $\times$ 19 layers) driven by meteorological fields from the NCEP reanalysis (Kalnay et al., 1996). NCEP is used since v4.1 again (rather than ERA-Interim) as only NCEP is currently available before 1980.

The *cost function minimization* uses the Conjugate Gradient algorithm, enhanced by a re-orthonormalization after each iteration to avoid the usual degradation of the convergence rate. The re-orthonormalization requires to store the state vectors and gradients of all iterations performed, which however opens the additional possibility to re-calculate the solution also for tighter prior constraints without the need to run the iterative minimization again. It also accumulates information about the a-posteriori covariance matrix, though the actual calculation of matrix elements generally needs further dedicated iterations.

### A2    The standard inversion s85oc_v4.1s

In comparison to the basic v4.1 runs (Sect. A1), the particular run s85oc_v4.1s involves 3 specifics or differences, respectively:

The station set s85v21 is used, comprising the 23 stations marked with * in Table 1.

The calculation is done over the shorter period 1980-2017 (indicated by the appended "s" in the version tag).

The entire *Ocean flux* (including interannual variations) is fixed to the CarboScope estimates oc_v1.5 (update of Rödenbeck et al., 2014) based on an interpolation of $pCO_2$ data from the SOCATv5 data base (Bakker et al., 2016). Fixed ocean fluxes are used here because atmospheric inversions are known to have limited capability to correctly assign signals to land or ocean (Peylin et al., 2013). While this error is relatively small for the land fluxes, it means a large relative error for the ocean fluxes, because the ocean variability is much smaller than the land variability. The $pCO_2$ data offer a much closer constraint on ocean $CO_2$ fluxes in well-observed regions (northern extratropics, tropical Pacific), and constrain at least some features (seasonality, decadal trends) in most ocean areas. (For the NEE-T inversion, fixed ocean fluxes are particularly beneficial because they avoid the need of time-dependent degrees of freedom.)

### Competing interests

The authors declare that they have no conflict of interest.

*Acknowledgements.* This study would not be possible without the sustained work of many colleagues involved in the measurement and distribution of atmospheric $CO_2$ data; we would like to thank for all their support. We are grateful to L. Aragão, P. Gentine, M. Jung, E. Kort, and M. Reichstein for inspiring discussions. We would like to thank the staff of the DKRZ supercomputing centre for their great support, in particular H. Bockelmann for optimizing the inversion and TM3 codes. We gratefully acknowledge that this work uses eddy covariance data acquired and shared by the FLUXNET community, including these networks: AmeriFlux,

AfriFlux, AsiaFlux, CarboAfrica, CarboEuropeIP, CarboItaly, CarboMont, ChinaFlux, Fluxnet-Canada, GreenGrass, ICOS, KoFlux, LBA, NECC, OzFlux-TERN, TCOS-Siberia, and USCCC. The FLUXNET eddy covariance data processing and harmonization was carried out by the European Fluxes Database Cluster, AmeriFlux Management Project, and Fluxdata project of FLUXNET, with the support of CDIAC and ICOS Ecosystem Thematic Center, and the OzFlux, ChinaFlux and AsiaFlux offices. This project was supported in part by US NSF, and National Aeronautics and Space Administration (NASA) under Grants 1304270 and NNX17AE74G. The service charges for this open access publication have been covered by the Max Planck Society.

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

**Table 1.** Atmospheric $CO_2$ measurement stations used in the NEE-T inversion. The smaller set of stations used in the standard inversion is labelled by an asterisk. The 8 parts individually omitted in sensitivity tests are separated by horizontal lines. Institutions are referenced by: AEMET: Gomez-Pelaez and Ramos (2011); BGC: Thompson et al. (2009); CSIRO: Francey et al. (2003); EC: Worthy (2003); FMI: Kilkki et al. (2015); HMS: Haszpra et al. (2001); IAFMS: Colombo and Santaguida (1994); JMA: Watanabe et al. (2000); LSCE: Monfray et al. (1996); NIES: Tohjima et al. (2008); NIPR: Morimoto et al. (2003); NOAA: Conway et al. (1994); Saitama: http://www.pref.saitama.lg.jp/b0508/cess-english/index.html; SAWS: Labuschagne et al. (2003); SIO: Keeling et al. (2005), Manning and Keeling (2006); UBA: Levin et al. (1995). Appended letters give record type: (f): flask data, mostly weekly; (h): in-situ data, mostly hourly; (d): in-situ data, day-time only; (n): in-situ data, night-time only.

| Code | Latitude (°) | Longitude (°) | Height (m a.s.l.) | Institution |
|------|------|------|------|------|
| *CMN | 44.18 | 10.70 | 2165 | IAFMS(n) |
| *LJO | 32.87 | -117.25 | 15 | SIO(f) |
| *ASC | -7.97 | -14.40 | 88 | NOAA(f) |
| *BHD | -41.40 | 174.90 | 85 | SIO(f) |
| *BRW | 71.32 | -156.61 | 13 | NOAA(h,f), SIO(f) |
| *CHR | 1.70 | -157.16 | 3 | NOAA(f) |
| *MID | 28.21 | -177.37 | 10 | NOAA(f) |
| *MLO | 19.53 | -155.57 | 3417 | NOAA(h,f), SIO(f) |
| *SPO | -89.97 | -24.80 | 2816 | NOAA(h,f), SIO(f) |
| *SYO | -69.00 | 39.58 | 29 | NIPR(h) |
| *KER | -29.03 | -177.15 | 2 | SIO(f) |
| ESP | 49.38 | -126.54 | 27 | CSIRO(f), EC(f) |
| MQA | -54.48 | 158.97 | 13 | CSIRO(f) |
| RYO | 39.03 | 141.83 | 230 | JMA(d) |
| MNM | 24.30 | 153.97 | 8 | JMA(d) |
| MHD | 53.32 | -9.81 | 18 | NOAA(f) |
| RPB | 13.16 | -59.43 | 19 | NOAA(f) |
| UTA | 39.90 | -113.72 | 1332 | NOAA(f) |
| HUN | 46.95 | 16.64 | 353 | HMI(d), NOAA(f) |
| AZR | 38.76 | -27.23 | 23 | NOAA(f) |
| HBA | -75.58 | -26.61 | 24 | NOAA(f) |
| LEF | 45.93 | -90.26 | 791 | NOAA(f) |
| SEY | -4.68 | 55.53 | 6 | NOAA(f) |
| CPT | -34.35 | 18.48 | 230 | SAWS(d) |
| PAL | 67.96 | 24.12 | 565 | FMI(d), NOAA(f) |
| WLG | 36.28 | 100.91 | 3852 | NOAA(f) |
| HAT | 24.05 | 123.80 | 10 | NIES(f) |
| SBL | 43.93 | -60.01 | 5 | EC(d,f) |
| CRZ | -46.43 | 51.85 | 202 | NOAA(f) |
| SGP | 36.71 | -97.49 | 348 | NOAA(f) |
| SUM | 72.60 | -38.42 | 3214 | NOAA(f) |
| WES | 54.93 | 8.32 | 12 | UBA(d) |
| AVI | 17.75 | -64.75 | 5 | NOAA(f) |
| EIC | -27.15 | -109.44 | 63 | NOAA(f) |
| ICE | 63.40 | -20.29 | 124 | NOAA(f) |
| TIK | 71.60 | 128.89 | 29 | NOAA(f) |
| CVR | 16.86 | -24.87 | 10 | BGC(f) |
| ZOT301 | 60.80 | 89.35 | 301 a.gr. | BGC(d,f) |
| POCN30 | 29.48 | -134.24 | 20 | NOAA(f) |
| POCN20 | 19.69 | -132.68 | 20 | NOAA(f) |
| POCN10 | 9.68 | -140.37 | 20 | NOAA(f) |
| POC000 | 0.60 | -150.35 | 20 | NOAA(f) |
| POCS10 | -10.02 | -3.61 | 20 | NOAA(f) |
| POCS20 | -20.28 | 0.08 | 20 | NOAA(f) |
| POCS30 | -29.68 | -0.04 | 20 | NOAA(f) |

| Code | Latitude (°) | Longitude (°) | Height (m a.s.l.) | Institution |
|------|------|------|------|------|
| *ALT | 82.47 | -62.42 | 202 | CSIRO(f), EC(f), NOAA(f) |
| *CBA | 55.21 | -162.71 | 41 | NOAA(f), SIO(f) |
| *CGO | -40.67 | 144.70 | 130 | CSIRO(f), NOAA(f) |
| *GMI | 13.39 | 144.66 | 6 | NOAA(f) |
| *IZO | 28.30 | -16.50 | 2367 | AEMET(h) |
| *KEY | 25.67 | -80.18 | 4 | NOAA(f) |
| *KUM | 19.51 | -154.82 | 22 | NOAA(f), SIO(f) |
| *NWR | 40.04 | -105.60 | 3526 | NOAA(f) |
| *PSA | -64.92 | -64.00 | 12 | NOAA(f), SIO(f) |
| *SHM | 52.72 | 174.11 | 27 | NOAA(f) |
| *SMO | -14.24 | -170.57 | 51 | NOAA(h,f), SIO(f) |
| *AMS | -37.80 | 77.54 | 55 | LSCE(d) |
| CFA | -19.28 | 147.06 | 5 | CSIRO(f) |
| MAA | -67.62 | 62.87 | 42 | CSIRO(f) |
| SIS | 60.18 | -1.26 | 31 | BGC(f), CSIRO(f) |
| SCH | 47.92 | 7.92 | 1205 | UBA(n) |
| BMW | 32.26 | -64.88 | 46 | NOAA(f) |
| TAP | 36.72 | 126.12 | 21 | NOAA(f) |
| UUM | 44.45 | 111.10 | 1012 | NOAA(f) |
| ASK | 23.26 | 5.63 | 2715 | NOAA(f) |
| TDF | -54.86 | -68.40 | 20 | NOAA(f) |
| WIS | 30.41 | 34.92 | 319 | NOAA(f) |
| ZEP | 78.91 | 11.89 | 479 | NOAA(f) |
| FSD | 49.88 | -81.57 | 250 | EC(d) |
| YON | 24.47 | 123.02 | 30 | JMA(d) |
| COI | 43.15 | 145.50 | 45 | NIES(f) |
| CYA | -66.28 | 110.52 | 55 | CSIRO(f) |
| THD | 41.04 | -124.15 | 112 | NOAA(f) |
| CIB | 41.81 | -4.93 | 848 | NOAA(f) |
| KZD | 44.26 | 76.22 | 506 | NOAA(f) |
| LLN | 23.47 | 120.87 | 2867 | NOAA(f) |
| NAT | -5.66 | -35.22 | 53 | NOAA(f) |
| NMB | -23.57 | 15.02 | 461 | NOAA(f) |
| STM | 66.00 | 2.00 | 3 | NOAA(f) |
| STP | 50.00 | 145.00 | 0 | SIO(f) |
| BIK300 | 53.22 | 23.02 | 300 a.gr. | BGC(f) |
| DDR | 36.00 | 139.18 | 840 | Saitama(n) |
| KEF+RYF | var. | var. | 0 | JMA(f) |
| POCN25 | 25.20 | -133.99 | 20 | NOAA(f) |
| POCN15 | 15.07 | -135.22 | 20 | NOAA(f) |
| POCN05 | 4.80 | -145.11 | 20 | NOAA(f) |
| POCS05 | -4.66 | -4.24 | 20 | NOAA(f) |
| POCS15 | -14.72 | -0.15 | 20 | NOAA(f) |
| POCS25 | -25.01 | -0.17 | 20 | NOAA(f) |

**Table 2.** Eddy covariance sites used for comparison. For vegetation type abbreviations, see Fig. 3 (caption)

| FLUXNET-ID | Data period | Latitude (°) | Longitude (°) | Vegetation type |
|---|---|---|---|---|
| AU-How | 2001–2014 | -12.4943 | 131.1523 | WSA |
| AU-Tum | 2001–2014 | -35.6566 | 148.1517 | EBF |
| BE-Bra | 1996–2014 | 51.3092 | 4.5206 | MF |
| BE-Vie | 1996–2014 | 50.3051 | 5.9981 | MF |
| CA-Man | 1994–2008 | 55.8796 | -98.4808 | ENF |
| CH-Dav | 1997–2014 | 46.8153 | 9.8559 | ENF |
| DE-Hai | 2000–2012 | 51.0792 | 10.4530 | DBF |
| DE-Tha | 1996–2014 | 50.9624 | 13.5652 | ENF |
| DK-Sor | 1996–2014 | 55.4859 | 11.6446 | DBF |
| DK-ZaH | 2000–2014 | 74.4732 | -20.5503 | GRA |
| FI-Hyy | 1996–2014 | 61.8474 | 24.2948 | ENF |
| FI-Sod | 2001–2014 | 67.3619 | 26.6378 | ENF |
| FR-LBr | 1996–2008 | 44.7171 | -0.7693 | ENF |
| FR-Pue | 2000–2014 | 43.7414 | 3.5958 | EBF |
| GF-Guy | 2004–2014 | 5.2788 | -52.9249 | EBF |
| IT-Col | 1996–2014 | 41.8494 | 13.5881 | DBF |
| IT-Cpz | 1997–2009 | 41.7052 | 12.3761 | EBF |
| IT-Lav | 2003–2014 | 45.9562 | 11.2813 | ENF |
| IT-Ren | 1998–2013 | 46.5869 | 11.4337 | ENF |
| IT-SRo | 1999–2012 | 43.7279 | 10.2844 | ENF |
| NL-Loo | 1996–2013 | 52.1666 | 5.7436 | ENF |
| RU-Cok | 2003–2014 | 70.8291 | 147.4943 | OSH |
| RU-Fyo | 1998–2014 | 56.4615 | 32.9221 | ENF |
| US-Ha1 | 1991–2012 | 42.5378 | -72.1715 | DBF |
| US-Los | 2000–2014 | 46.0827 | -89.9792 | WET |
| US-Me2 | 2002–2014 | 44.4523 | -121.5574 | ENF |
| US-MMS | 1999–2014 | 39.3232 | -86.4131 | DBF |
| US-NR1 | 1998–2014 | 40.0329 | -105.5464 | ENF |
| US-PFa | 1995–2014 | 45.9459 | -90.2723 | MF |
| US-Syv | 2001–2014 | 46.2420 | -89.3477 | MF |
| US-Ton | 2001–2014 | 38.4316 | -120.9660 | WSA |
| US-UMB | 2000–2014 | 45.5598 | -84.7138 | DBF |
| US-Var | 2000–2014 | 38.4133 | -120.9507 | GRA |
| US-WCr | 1999–2014 | 45.8059 | -90.0799 | DBF |
| ZA-Kru | 2000–2010 | -25.0197 | 31.4969 | SAV |

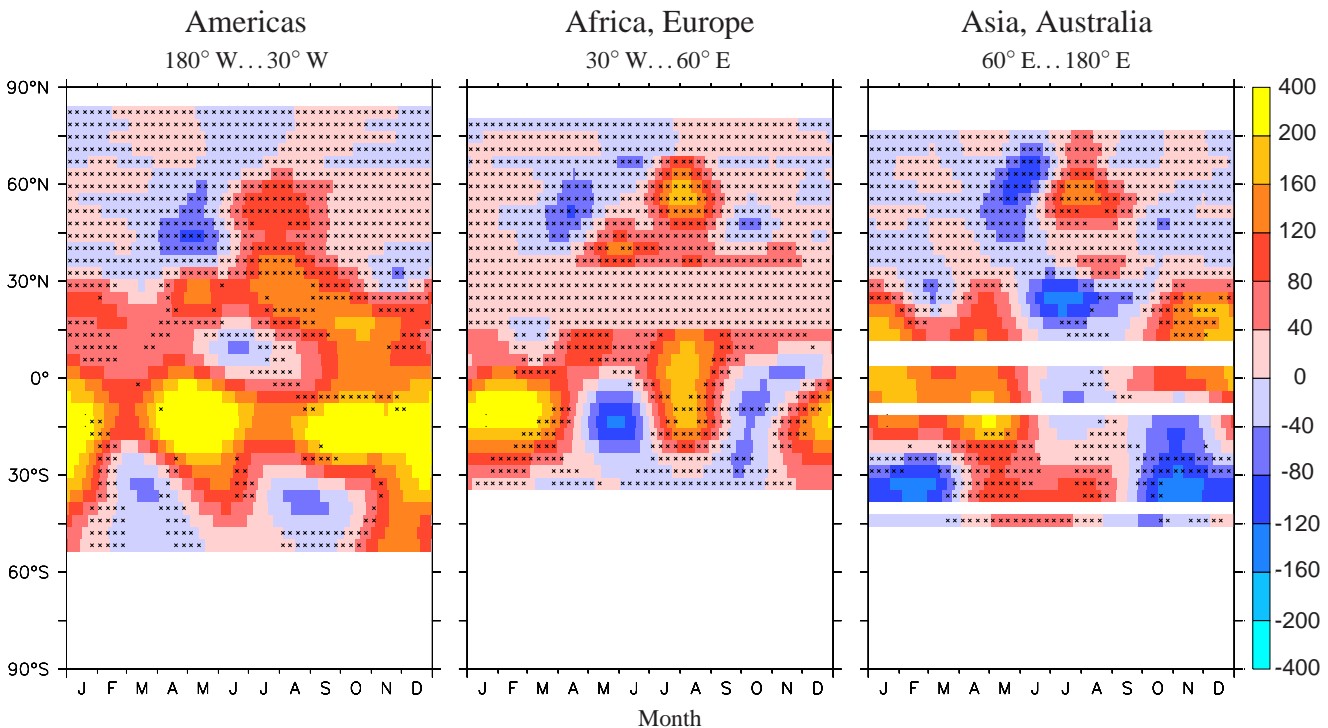

**Figure 1.** "Interannual climate sensitivity" $\gamma_{\mathrm{NEE\text{-}T}}$ in $(\mathrm{gC/m^{-2}/yr})/\mathrm{K}$ shown as Hovmöller diagrams: Longitudinal averages of $\gamma_{\mathrm{NEE\text{-}T}}$ are plotted as color over latitude (vertical) and month of the year (horizontal). The stippling indicates robustness: crosses mark values with absolute deviations $\leq 40\,(\mathrm{gC/m^{-2}/yr})/\mathrm{K}$ (1 color level) of all sensitivity cases from the base case.

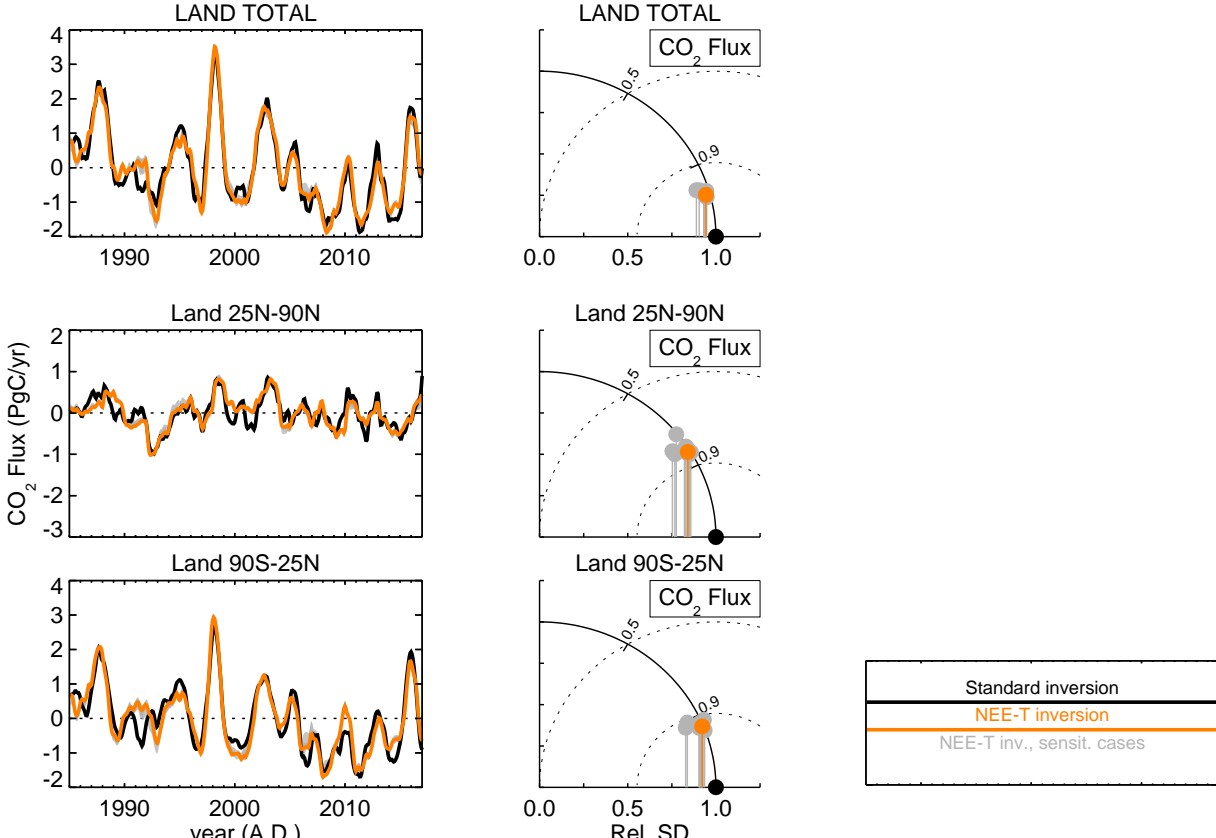

**Figure 2.** Left: Interannual anomalies of NEE integrated over all land (top), northern extratropical land (middle), and tropical plus southern land (bottom), as estimated by the standard inversion (Sect. 2.1, black) and different runs of the NEE-T inversion (Sect. 2.2, orange). The gray band comprises the results of the sensitivity cases. Right: Taylor diagrams quantifying the agreement between the NEE-T inversions and the standard inversion. Due to the construction of the Taylor diagram (Taylor, 2001), the horizontal position of a point gives the relative fraction of the reference signal present in the test time series, while the vertical distance of this point from the horizontal axis gives the relative amplitude (temporal standard deviation) of any additional signal components uncorrelated to the reference signal.

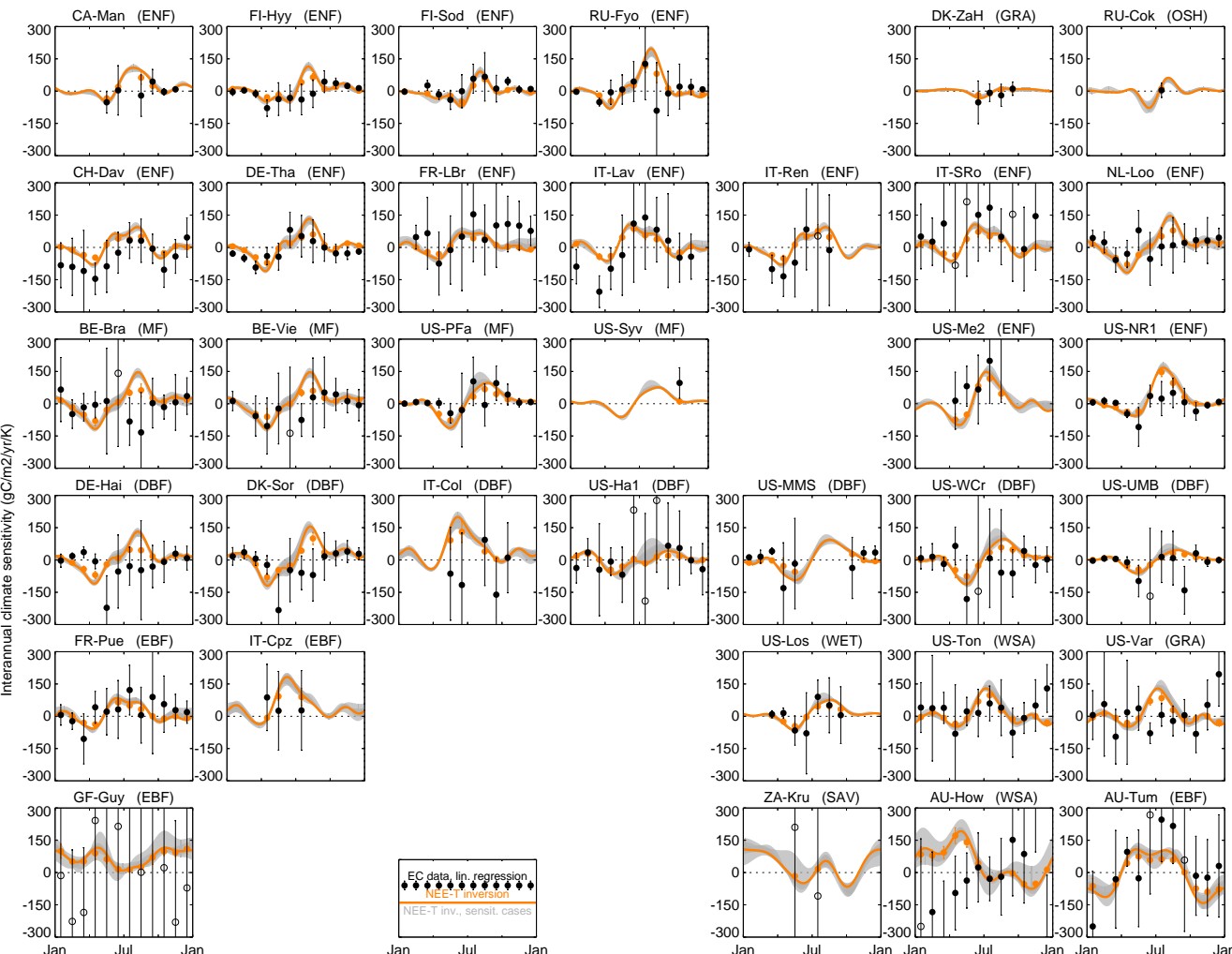

**Figure 3.** Comparison between the "interannual climate sensitivities" calculated from the inversion and from eddy covariance (EC) data, for various sites with longer EC records. Black dots give the sensitivities $g_{NEE-T}^{EC}$ calculated by linear regression of monthly EC $CO_2$ flux data (FLUXNET2015 data set) against monthly air temperature co-measured at the flux towers (months with data in only 6 years or less are discarded). The error bars around the dots comprise the confidence intervals of the regression slopes (at the 90% confidence level); if the confidence interval is above $300\,(gC/m^{-2}/yr)/K$ (i.e., larger than the typical seasonal range), the corresponding dot is hollow. Orange and gray lines give the sensitivities $\gamma_{NEE-T}$ taken directly from various NEE-T inversions (base and sensitivity cases as in Fig. 2) at the respective pixels enclosing the EC site locations. To allow a more direct comparison between NEE-T inversion results and EC data, sensitivities for the inversion (base case) have also been calculated by linear regression from the total monthly-mean non-fossil $CO_2$ flux and the temperature field employed in the inversions, in the same way and subsampled at the same months as for the EC data; these $g_{NEE-T}^{Inv}$ are shown as orange dots. Panels are roughly ordered by latitude and land cover type (DBF: Decidious broadleaf forest, EBF: Evergreen broadleaf forest, ENF: Evergreen needleleaf forest, GRA: Grassland, MF: Mixed forest, OSH: Open Shrubland, SAV: Savanna, WET: Permanent wetland, WSA: Woody Savanna). See Table 2 for EC site locations.

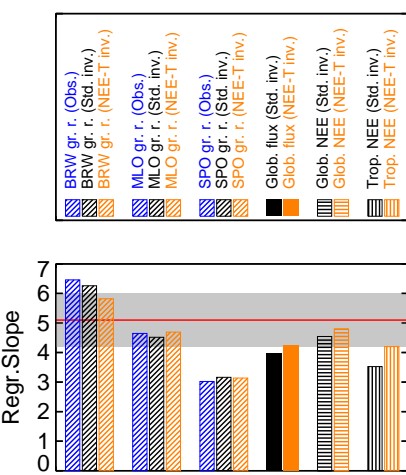

**Figure 4.** Effective large-scale interannual climate sensitivities $(PgC\,yr^{-1}K^{-1})$ calculated from the standard inversion (black), from the NEE-T inversion (orange), or from observed atmospheric $CO_2$ (blue). The sensitivities refers to interannual variations in the $CO_2$ growth rate at 3 selected atmospheric stations (Point Barrow, Alaska (BRW), Mauna Loa, Hawaii (MLO) and South Pole (SPO), diagonally hashed), in the global total $CO_2$ exchange (solid bars), in the global terrestrial NEE (horizontally hashed), or in tropical NEE $(25^\circ$ N–$90^\circ$ S, vertically hashed), all regressed against interannual variations in air temperature averaged across tropical land $(25^\circ$ N– $25^\circ$ S) over 1985-2016. The red line surrounded by gray shading denotes the result $5.1 \pm 0.9\,PgC\,yr^{-1}K^{-1}$ by Cox et al. (2013), even though calculated in a slightly different way.