# Peer review of "How does the terrestrial carbon exchange respond to interannual climatic variations? A quantification based on atmospheric ${\rm CO_2}$ data"

_Biogeosciences, 2018_

## Referee Comment (RC1) · Anonymous Referee #1 · 8 Feb 2018

This paper quantified the sensitivity of the terrestrial carbon exchange to interannual climatic variations using a new formulation of atmospheric CO2 flux inversion. Instead of optimizing terrestrial carbon exchange directly as in classical CO2 flux inversion, this study optimized the sensitivity of terrestrial carbon exchange interannual variability (IAV) to temperature, which itself has been used as an emergent quantity to constrain the predictions of future terrestrial biosphere carbon accumulations. They found that the sensitivity changes with latitudes and seasons. The results over the NH extratropics are more robust and agree better with independent sensitivity calculated from eddy covariance observations from flux towers. The paper is well written, and the proposed method is quite interesting. I recommend this paper for publication after minor revision.

[Figure]

Here are my detailed comments:

1. I would recommend adding more details about the inversion system. In describing the standard inversions in the first paragraph in section 2.1, it would be easier for readers to follow if they can add a cost function equation, and then describe how they define each term in that cost function. Currently, it is hard to understand the sentence: "The cost function additionally brings in a-priori information to regularize the estimation, in particular spatial and temporal smoothness constrains on the flux field". I have to guess from that the authors are talking about the prior error covariance structure.

2. The details of sensitivity experiments described in section 2.3 are lacking, which make it hard to judge whether the uncertainties calculated from these sensitivity experiments are realistic. It would be helpful to describe the first three sensitivity experiments quantitatively. How much longer are the spatial correlations and temporal correlations in the sensitivity experiments? and how much have the a priori uncertainties been reduced in these sensitivity experiments?

3. Besides the sensitivity of terrestrial carbon exchange (NEE) IAV, the atmospheric $CO_2$ inversion described in this study also optimized the long-term trend and the seasonality trend of NEE. Since only $CO_2$ observations were assimilated, the sum of these three terms should agree with observed $CO_2$. Therefore, these three quantities mathematically have intrinsic correlations. It would be helpful to discuss the dependency of the estimated IAV sensitivity to the a priori assumptions of the other two terms in equation (2). Also, I would suggest adding a few sentences discussing whether the estimated trends are realistic, though this study focuses on the sensitivity of IAV.

4. Figure 1 used 40 gC/m2/yr as a threshold for robustness of the calculated sensitivity. Where did this number come from? What is the basis?

5. The unit in Figure 1 should be gC/m2/yr.

6. In the first paragraph in the introduction, "the response of NEE on. . ." should be "the

response of NEE to. . .”
* * *
Interactive
comment

---

## Referee Comment (RC2) · Anonymous Referee #2 · 25 Feb 2018

The authors present a method to estimate climate relevant parameters from the atmospheric long-term records of CO2. These parameters describe the regression of surface CO2 exchange (NEE) onto temperature anomalies (T), a metric that has previously been assessed from a suite of climate models and from atmospheric observations. The inversion method to derive these metrics is based on a well-documented and exhaustively tested system, which also offers the authors the opportunity to benchmark their new inversion method to the existing one. Their findings show that the sensitivity of NEE to temperature anomalies follows seasonal patterns on the NH which are as one would expect based on limitations of (light) and temperature, while in other areas there is not enough data to constrain the seasonal cycle well. On interannual time

scales, both the standard inversion and the gamma-based inversion give very similar temporal patterns of NEE, giving further confidence to the ability of this new type of inversion to use temperature patterns and the derived climate sensitivity to T (as a proxy for other processes) to constrain NEE variations.

I overall find the paper well written, and interesting, and it opens up a new approach to inverse modeling of CO2 mixing ratios. The paper is perhaps a little bit thin on "new findings" but given the novelty of the method and the actual importance of benchmarking well, this does not diminish the value of the paper for me. I would like to see some further details provided about the method since it takes an important part of the value of the paper. Besides some few additional considerations detailed below, the paper should be ready for publication after some minor revisions. I am sure it will make an interesting paper for the readers of this journal.

Minor comments

What I am surprised to \*not\* find in this paper is the context of gamma as given by the Cox et al (2013) constraint on climate sensitivity. As a simple step, I wonder if you can post-aggregate your monthly gridbox results to global yearly values. This would have to give a number comparable to the Cox et al., (2013) values(5.1 $\pm$ 0.9 GtC yr-1 K-1, but also found in a number of preceding and subsequent papers on gamma)? And would creating an annual number yield more robust tropical results when it comes to the sensitivity runs, or do they remain scattered? Finally, it seems to me you are also one of the first who can make a gamma for the non-tropical regions which was not something that Cox et al could do from the global growth rate analysis (they did not have extra-tropical-only growth rates like the CO2 records you use). Comparing this gamma to the tropical one, and setting it out as a target for TBMs could be a nice addition to this study. Please comment on the feasibility of this, and if you agree it can be done, please add this analysis to the results and discussion.

Point-by-point comments

- Section 2.1: I would like to see a brief summary of the number of spatial and temporal unknowns that is associated with each term in equations (1) and (2)

- page 3, line 33, the meaning of the $\Delta NEE/\Delta T$ term is not quite clear. What do the $\Delta$'s refer to in this equation? Why is it presented at this point?

- Page 4, line 8: I do not see these residual terms in the equation? Unless you refer to the Trend and SC components? But these are only long-term (slow) changes and therefore cannot be expected to prevent spurious changes in gamma, unless I misunderstand what you mean to say here?

- Page 4, line 17: "For each degree of freedom (Fourier mode)..." this formulation is not clear to me, as I have not read about a Fourier decomposition and cannot relate it to the degrees of freedom. Hence my request to improve the description of the temporal components of the system in Eq 2.

- Page, line 23: "...stay in the data residual of the inversion." Why are you so sure that they will end up there, and not aliased into one of the parameters?

- Page 4, line 28: "such that any data point influences all years of the calculation period simultaneously." This suggests, like the first discussion of the results, that for each gridbox, 12 gamma parameters are estimated representing the sensitivity in each month. This sensitivity is repeated for each year of the analysis period. Correct? If so, I advise to make this clearer from the descriptions near Eq 2.

Page 5, line 23: Were these regression lines forced to go through (0,0), or was an offset also fit?

Page 6, line 10: Did you mean to write "representation"?

Page 6, line 13: after mentioning in the previous paragraph that you will interpret gamma more broadly as an interannual climate sensitivity, you here immediately fall back on the temperature limitation of spring NEE. But would a light limitation not also be just a good an explanation as per your own reasoning? In that line, spring conditions

with higher T would also have higher incoming solar radiation which would stimulate photosynthesis. Please consider this hypothesis and see if it can find a place in the text if you feel it has merit.

Page 9, line 5: "The NEE-T regression is an example that derived relationships are able to brigde this scale gap" Typo in "bridge", but also, I suggest a statement that this might not necessarily be true in the tropical regions where gamma might be most relevant, but no EC data was available to confirm the inverse results.

Page 9, line 10: "pan-topical". Typo

Page 9, line 18, "impressively" I am not sure this is the qualification you wanted to insert here. If so, what is so impressive here?

Page 10, line 26: "forcast". Typo, 2x in sentence

Page 11, line 23: "outlieres" typo

Page 11, line 28: "insufficiencies". I suggest "deficiencies" or "inabilities"

Page 11, lines 29 and 30: "anomlies" typo

Figure 3: I find the presentation of these results really very dense, and found it hard to summarize for myself the meaning from all the panels. Could this figure be improved by presenting some statistical summary of the results per PFT? Or a simple temporal correlation coefficient per site perhaps. After all, given the large error bars the temporal patterns are mostly guiding the eye anyway in these panels.

---

## Author Comment (AC1) · 15 Mar 2018

**Authors' Response to Anonymous Reviewer 1**

*This paper quantified the sensitivity of the terrestrial carbon exchange to interannual climatic variations using a new formulation of atmospheric CO2 flux inversion. Instead of optimizing terrestrial carbon exchange directly as in classical CO2 flux inversion, this study optimized the sensitivity of terrestrial carbon exchange interannual variability (IAV) to temperature, which itself has been used as an emergent quantity to constrain the predictions of future terrestrial biosphere carbon accumulations. They found that the sensitivity changes with latitudes and seasons. The results over the NH extratropics*

*are more robust and agree better with independent sensitivity calculated from eddy covariance observations from flux towers. The paper is well written, and the proposed method is quite interesting. I recommend this paper for publication after minor revision.*

We would like to thank the Reviewer for taking the time to review our manuscript and for her/his helpful and supportive comments.

*Here are my detailed comments: 1. I would recommend adding more details about the inversion system. In describing the standard inversions in the first paragraph in section 2.1, it would be easier for readers to follow if they can add a cost function equation, and then describe how they define each term in that cost function. Currently, it is hard to understand the sentence: "The cost function additionally brings in a-priori information to regularize the estimation, in particular spatial and temporal smoothness constrains on the flux field". I have to guess from that the authors are talking about the prior error covariance structure.*

We agree that the sentence cited by the Reviewer is not clear. Actually, Sect. 2.1 was not intended to give a detailed description of the pre-existing inversion algorithm (because that would be very long), but rather to refer the reader to the specific items in the Appendix or the complete description in the Technical Report Rödenbeck (2005). We feel that it would be better not to overload Sect. 2.1 with technical details. However, we will extend the Appendix and give more information there.

We reformulated the unclear sentence and the previous one into: "...closest match between observed and simulated $CO_2$ mole fractions. In addition, the estimation is regularized by a-priori constraints meant to suppress excessive spatial and high-frequency variability in the flux field."

*2. The details of sensitivity experiments described in section 2.3 are lacking, which make it hard to judge whether the uncertainties calculated from these sensitivity experiments are realistic. It would be helpful to describe the first three sensitivity experiments quantitatively. How much longer are the spatial correlations and temporal correlations*

*in the sensitivity experiments? and how much have the a priori uncertainties been reduced in these sensitivity experiments?*

We added the quantitative information for cases (1) to (3).

*3. Besides the sensitivity of terrestrial carbon exchange (NEE) IAV, the atmospheric CO2 inversion described in this study also optimized the long-term trend and the seasonality trend of NEE. Since only CO2 observations were assimilated, the sum of these three terms should agree with observed CO2. Therefore, these three quantities mathematically have intrinsic correlations. It would be helpful to discuss the dependency of the estimated IAV sensitivity to the a priori assumptions of the other two terms in equation (2). Also, I would suggest adding a few sentences discussing whether the estimated trends are realistic, though this study focuses on the sensitivity of IAV.*

Though long-term, seasonal, and interannual degrees of freedom are indeed linked theoretically, the actual a-posteriori correlations are very small (see, e.g., Fig 16 of the Technical Report Rödenbeck, 2005). In sensitivity tests during the development of the NEE-T inversion, we did not find any strong dependence between these time scales either. For the manuscript, these sensitivity tests were not selected because of their small effect.

In response to this comment, we added another paragraph to Sect. 4.1: "Besides the interannual variations, the NEE-T inversion also reproduces the small negative trend in NEE through its residual term $f^{\text{adj}}_{\text{NEE,Trend}}$ in Eq. (2) (Fig. 2). Likewise, it reproduces the northern extratropical increase in seasonal cycle amplitude through its residual term $f^{\text{adj}}_{\text{NEE,SCTrend}}$ (not shown)."

*4. Figure 1 used 40 gC/m2/yr as a threshold for robustness of the calculated sensitivity. Where did this number come from? What is the basis?*

Unfortunately, we did not find a truly objective criterion for robustness. We therefore took a threshold that would be some meaningful fraction of the size of the structures

seen in Fig 1, and visualize the qualitative difference between extratropics and trop-ics. The threshold of 40 (gC/m2/yr)/K corresponds to 1 step of the color scale which seemed to fulfill these intentions.

*5. The unit in Figure 1 should be gC/m2/yr.*

Fig 1 shows the sensitivity gamma = dNEE/dT, which has units of flux per Kelvin, here (gC/m2/yr)/K.

*6. In the first paragraph in the introduction, "the response of NEE on..." should be "the response of NEE to..."*

Thank you for spotting this. Corrected

---

## Author Comment (AC2) · 15 Mar 2018

**Authors' Response to Anonymous Reviewer 2**

*The authors present a method to estimate climate relevant parameters from the atmospheric long-term records of CO2. These parameters describe the regression of surface CO2 exchange (NEE) onto temperature anomalies (T), a metric that has previously been assessed from a suite of climate models and from atmospheric observations. The inversion method to derive these metrics is based on a well-documented and exhaustively tested system, which also offers the authors the opportunity to benchmark their new inversion method to the existing one. Their findings show that the sensitivity*

*of NEE to temperature anomalies follows seasonal patterns on the NH which are as one would expect based on limitations of (light) and temperature, while in other areas there is not enough data to constrain the seasonal cycle well. On interannual time scales, both the standard inversion and the gamma-based inversion give very similar temporal patterns of NEE, giving further confidence to the ability of this new type of inversion to use temperature patterns and the derived climate sensitivity to T (as a proxy for other processes) to constrain NEE variations. I overall find the paper well written, and interesting, and it opens up a new approach to inverse modeling of CO2 mixing ratios. The paper is perhaps a little bit thin on "new findings" but given the novelty of the method and the actual importance of benchmarking well, this does not diminish the value of the paper for me. I would like to see some further details provided about the method since it takes an important part of the value of the paper. Besides some few additional considerations detailed below, the paper should be ready for publication after some minor revisions. I am sure it will make an interesting paper for the readers of this journal.*

We would like to thank the Reviewer for taking the time to review our manuscript and for her/his helpful and supportive comments.

*Minor comments*

*What I am surprised to **not** find in this paper is the context of gamma as given by the Cox et al (2013) constraint on climate sensitivity. As a simple step, I wonder if you can post-aggregate your monthly gridbox results to global yearly values. This would have to give a number comparable to the Cox et al., (2013) values (5.1 ± 0.9 GtC yr-1 K-1, but also found in a number of preceding and subsequent papers on gamma)? And would creating an annual number yield more robust tropical results when it comes to the sensitivity runs, or do they remain scattered? Finally, it seems to me you are also one of the first who can make a gamma for the non-tropical regions which was not something that Cox et al could do from the global growth rate analysis (they did not have extra-tropical-only growth rates like the CO2 records you use). Comparing*

*this gamma to the tropical one, and setting it out as a target for TBMs could be a nice addition to this study. Please comment on the feasibility of this, and if you agree it can be done, please add this analysis to the results and discussion.*

The data-based estimates of the spatially and seasonally resolved $\gamma_{NEE-T}$ presented in the manuscript can directly be employed as a benchmark of terrestrial biosphere models or Earth system models, by regressing the simulated NEE against temperature in the same way and comparing to the data-based values. We realize from the Reviewer's comment that we did not formulate this clearly enough in the manuscript. We will therefore add a new subsection to the Discussion, explaining the application of the results as a model benchmark and their relation to effective global climate sensitivities as that by Cox et al (2013).

*Point-by-point comments*

*- Section 2.1: I would like to see a brief summary of the number of spatial and temporal unknowns that is associated with each term in equations (1) and (2)*

We assume that the background of this suggestion is to get an idea by how much the number of unknowns is reduced in the NEE-T inversion compared with the standard inversion. We agree that this was a missing piece of information, and added the following sentence to the 2nd paragraph of Sect 2.2: "By having only 13 degrees of freedom in the time dimension, the introduction of the regression term also regularizes the inversion further compared with the explicit interannual term of the standard inversion, which has 796 degrees of freedom in the time dimension."

*- page 3, line 33, the meaning of the dNEE/dT term is not quite clear. What do the delta's refer to in this equation? Why is it presented at this point?*

This term was meant to point out that the gamma values formally represent sensitivities of NEE to interannual temperature variations. As it seems to be confusing, we deleted this term, given that the beginning of Sect 3.1 introduces the meaning of gamma anyway. Instead, we added "(the NEE-T regression coefficients)" which may be a more helpful explanation at this point.

*- Page 4, line 8: I do not see these residual terms in the equation? Unless you refer to the Trend and SC components? But these are only long-term (slow) changes and therefore cannot be expected to prevent spurious changes in gamma, unless I misunderstand what you mean to say here?*

The expression "residual terms" indeed refers to the terms in the second line of Eq (2). They are residuals in the sense that they belong to the interannual variations but cannot be represented by the regression term as the main term of the NEE-T inversion. While the linear trend indeed may not interact with the regression term (since T has been detrended), the T field still contains some change in seasonal cycle which could exert some undesired influence on the estimated gamma values.

We added "(2nd line)" and converted the paragraphs into bullet points corresponding to the individual terms, to make the meaning of "residual terms" more clear. At this occasion, we noticed that we missed to mention the first of the residual terms, and added another bullet point (put as first one) for it.

*- Page 4, line 17: "For each degree of freedom (Fourier mode)..." this formulation is not clear to me, as I have not read about a Fourier decomposition and cannot relate it to the degrees of freedom. Hence my request to improve the description of the temporal components of the system in Eq 2.*

For practical reasons, the temporal variations in all adjustable terms used here (and in other CarboScope set-ups) are implemented as Fourier series. Temporal correlations can then simply be implemented by downweighting the Fourier modes of the higher frequencies according to the spectrum corresponding to the desired autocorrelation function. This is described in detail in the Technical Report Rödenbeck (2005). We will re-organize and enlarge the Appendix A to give at least the ideas of these implementation details.

The additional seasonal trend term $f_{NEE,SCTrend}$ could be implemented in the described way no matter whether $f_{NEE,Seas}$ was implemented in the Fourier domain or in the time domain. In order not to overload Sect 2.2 with material not essential to the application of the NEE-T inversion, we thus deleted "(Fourier mode)" rather than adding some explanation here.

We also corrected "$f_{NEE,Trend}$" into "$f_{NEE,SCTrend}$" on line 18, and added "the additional term".

*- Page, line 23: "...stay in the data residual of the inversion." Why are you so sure that they will end up there, and not aliased into one of the parameters?*

Our formulation may indeed be too optimistic. We reformulated into "Any further residual modes of variability (...) are not explicitly accounted for, as we lack sufficient a-priori information to model them explicitly. To the extend that they are uncorrelated to T variations, they will stay in the data residual of the inversion."

*- Page 4, line 28: "such that any data point influences all years of the calculation period simultaneously." This suggests, like the first discussion of the results, that for each gridbox, 12 gamma parameters are estimated representing the sensitivity in each month. This sensitivity is repeated for each year of the analysis period. Correct? If so, I advise to make this clearer from the descriptions near Eq 2.*

We enlarged the paranthesis on page 4 line 1–2 into "(with a correlation lengths of about 3 weeks, such that gamma contains 13 independent degrees of freedom in time, repeated every year)". We then also enlarged (and corrected, resp.) the second paranthesis (page 4 line 2) into "(with a correlation lengths of about 1600km in longitude direction and 800km in latitude direction, imposing a spatial smoothing on gamma over the same spatial scales as the smoothing imposed on the interannual flux anomalies $f_{NEE,IAV}$ in the standard inversion)"

*Page 5, line 23: Were these regression lines forced to go through (0,0), or was an*

*offset also fit?*

The fit also included an offset, as the slope is meant to refer to the anomalies.

*Page 6, line 10: Did you mean to write "representation"?*

Oops, yes. Corrected

*Page 6, line 13: after mentioning in the previous paragraph that you will interpret gamma more broadly as an interannual climate sensitivity, you here immediately fall back on the temperature limitation of spring NEE. But would a light limitation not also be just a good an explanation as per your own reasoning? In that line, spring conditions with higher T would also have higher incoming solar radiation which would stimulate photosynthesis. Please consider this hypothesis and see if it can find a place in the text if you feel it has merit.*

Thank you for pointing out this further possible mechanism. We added the sentence "Warmer conditions tend to coincide with higher incoming solar radiation in May and/or June in the northern extratropics (according to a correlation analysis of CRUNCEPv7 data, not shown), which would tend to amplify the direct temperature effect."

*Page 9, line 5: "The NEE-T regression is an example that derived relationships are able to brigde this scale gap" Typo in "bridge", but also, I suggest a statement that this might not necessarily be true in the tropical regions where gamma might be most relevant, but no EC data was available to confirm the inverse results.*

The sentence cited by the Reviewer was actually meant to point out the contrast between comparing fluxes and comparing derived relationships. We reformulated it to "In contrast to the fluxes, however, derived relationships (such as the NEE-T relationships considered here) may well be able to bridge this scale gap."

*Page 9, line 10: "pan-topical". Typo*

Corrected

*Page 9, line 18, "impressively" I am not sure this is the qualification you wanted to insert here. If so, what is so impressive here?*

"impressively" was deleted.

*Page 10, line 26: "forcast". Typo, 2x in sentence*

Corrected

*Page 11, line 23: "outlieres" typo*

Corrected

*Page 11, line 28: "insufficiencies". I suggest "deficiencies" or "inabilities"*

Changed to "deficiencies"

*Page 11, lines 29 and 30: "anomlies" typo*

Corrected

*Figure 3: I find the presentation of these results really very dense, and found it hard to summarize for myself the meaning from all the panels. Could this figure be improved by presenting some statistical summary of the results per PFT? Or a simple temporal correlation coefficient per site perhaps. After all, given the large error bars the temporal patterns are mostly guiding the eye anyway in these panels.*

Our intention was to not "hide" any information, and thus showed many EC sites, not only selected ones. We agree that this resulted in a somewhat heavy figure. However, we did not find a more compressed way of representation. The temporal correlation coefficients, as suggested by the Reviewer, are mostly not very high, because a mismatch in 1 or 2 out of 12 months (which happens at several sites in summer) destroyes the correlation. This does not mean, however, that the match was as bad in other parts of the year. We are therefore inclined to leave the figure in this state despite its complexity.

---

## Author Response (AR1)

Dear Christoph,

thank you very much for your decision. In the revised manuscript, we have changed all the points described in the answers to the referees.

The additional section (Sect 4.3) got a bit longer than percieved earlier and also has an additional figure (Fig 4), because we felt that this was needed to consider this issue properly.

Also the enlargement of the technical appendix now comprises more information than directly requested by the referees. However, given that the pieces of information that the referees had suggested to add are basic information, we felt it may be appropriate to give a rough but more or less complete overview of the algorithm in the appendix.

At the end of Sect 4.2, we made an additional change not yet described in the answers to the referees, by adding another sentence referring to Clark et al (2013) which we got aware of only recently.

Best regards
-Christian